# MCTBench: Multimodal Cognition towards Text-Rich Visual Scenes Benchmark

## Abstract

The comprehension of text-rich visual scenes has become a focal point for evaluating Multi-modal Large Language Models (MLLMs) due to their widespread applications. Current benchmarks tailored to the scenario emphasize perceptual capabilities, while overlooking the assessment of cognitive abilities. To address this limitation, we introduce a **M**ultimodal benchmark towards **T**ext-rich visual scenes, to evaluate the **C**ognitive capabilities of MLLMs through visual reasoning and content-creation tasks (**MCTBench**). To mitigate potential evaluation bias from the varying distributions of datasets, MCTBench incorporates several perception tasks (e.g., scene text recognition) to ensure a consistent comparison of both the cognitive and perceptual capabilities of MLLMs. To improve the efficiency and fairness of content-creation evaluation, we conduct an automatic evaluation pipeline. Evaluations of various MLLMs on MCTBench reveal that, despite their impressive perceptual capabilities, their cognition abilities require enhancement. We hope MCTBench will offer the community an efficient resource to explore and enhance cognitive capabilities towards text-rich visual scenes.

## 1 Introduction

Multimodal Large Language Models (MLLMs) OpenAI (2023); Team et al. (2023); Liu et al. (2023b); Li et al. (2024b) have exhibited promising performance across various cross-modal tasks, and revealed potential for widespread real-world applications. In practical applications, many images contain crucial textual elements that are essential for addressing specific challenges, such as key information extraction from receipts. Consequently, the ability to comprehend text-rich visual scenes can significantly enhance the practicality of MLLMs and drive innovative applications across multiple domains.

Recent benchmarks Liu et al. (2024c); Li et al. (2024a); iang Yue et al. (2023) have increasingly focused on evaluating MLLMs towards text-rich visual scenes. Nonetheless, the benchmarks are centred around evaluating perceptual capabilities yet overlook the assessment of cognitive abilities, which are a significant strength of MLLMs (as illustrated in Figure 1).

In this paper, we propose a **M**ultimodal benchmark to evaluate the **C**ognitive capabilities of MLLMs in **T**ext-rich visual scenes (**MCTBench**). To assess the cognitive abilities of MLLMs thoroughly, we design two types of tasks in the MCTBench: reasoning tasks for comprehension of the input scenes, and open-ended content-creation tasks for generating output responses. Besides, MCTBench integrates various perception tasks to study the differences with cognition tasks, while avoiding evaluation biases from varying dataset distributions. Fundamentally, MCTBench curates approximately 5.2k text-rich images from a wide range of public datasets, along with 8.5k rigorously annotated question-answer pairs categorized into three tasks: perception, reasoning and content-creation. The perception and reasoning tasks are formatted as multiple-choice questions for convenient evaluation, following common practices in Fu et al. (2024); Liu et al. (2024b); Li et al. (2023b). Due to the subjectivity and high cost of human evaluation in open-ended content creation, we establish an automated evaluation pipeline by leveraging sophisticated MLLMs (e.g., GPT-4V) as the evaluator, to compare the predictions of models against the provided references. Our experimental results demonstrate that MLLMs exhibit notably lower performance of cognition capabilities compared to perception in text-rich visual scenes, particularly for text-enhanced models. Furthermore, performances in cognition tasks (reasoning and content-creation) are improved with larger parameter scales. Our main

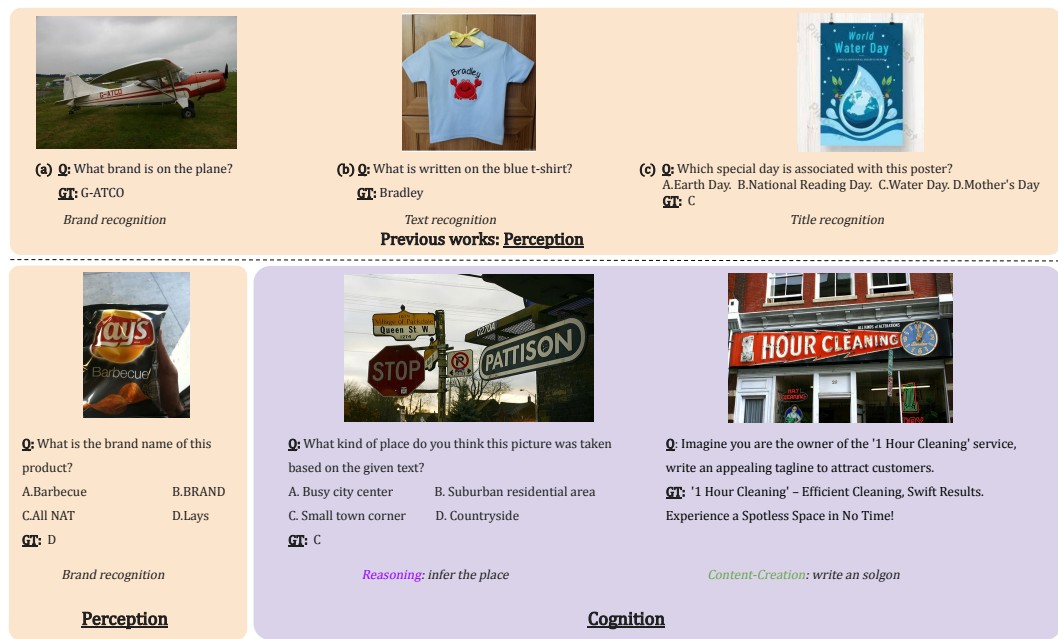

Figure 1: The Comparison between previous Benchmarks Singh et al. (2019); Liu et al. (2024c); Li et al. (2024a), and our proposed MCTBench. **Q** and **GT** stand for question and ground truth.

contributions are summarized as follows:

1. We propose a brand-new and large-scale benchmark for evaluating the cognitive capability of MLLMs towards text-rich visual scenes.

2. The evaluation on MCTBench highlights that MLLMs necessitate enhancements in their cognitive capabilities in text-rich visual scenes.

3. We develop an automated evaluation pipeline for the content-creation task, offering researchers an efficient tool for further investigation of cognitive capabilities.

## 2 RELATED WORK

### 2.1 MULTIMODAL LARGE LANGUAGE MODELS

The significant advancements in Large Language Models (LLMs) OpenAI (2023); Touvron et al. (2023); Chiang et al. (2023) have paved the way for recent research Team et al. (2023); Bai et al. (2023); Liu et al. (2023b); Chen et al. (2023a); Dai et al. (2023) into developing Multimodal Large Language Models (MLLMs) that integrate visual capabilities. Early works in this field Alayrac et al. (2022); Li et al. (2023c); Liu et al. (2023b); Chen et al. (2023a) have introduced various vision-language projectors such as Q-formerDai et al. (2023), Multi-Layer Perceptron (MLP) Liu et al. (2023b), and PerceiverAlayrac et al. (2022), which act as intermediaries between LLMs and visual encoders. Furthermore, these efforts have also established robust training paradigms for MLLMs. Building upon these foundational paradigms, recent initiatives Chen et al. (2023b); Lu et al. (2024a); Liu et al. (2023a); McKinzie et al. (2024) have focused on scaling the quality of training data, to enhance general visual capabilities effectively.

A primary challenge in the recent development of MLLMs is attaining fine-grained comprehension, exemplified by tasks such as Visual Question Answering (VQA) on text-rich images. To address this issue, increasing the resolution and integrating fine-grained visual features have been proven effective across various studies Feng et al. (2023); Liu et al. (2024d;a); Hu et al. (2023); Ye et al. (2023b;a); Li et al. (2024c); Wei et al. (2023). Additionally, works such as Feng et al. (2023); Hu et al. (2023);

| Benchmark | Text-Rich Oriented | #Image | #QAs | Perception | Reasoning | Content Creation | Answer Type |
|---|---|---|---|---|---|---|---|
| MME Fu et al. (2024) | ✘ | 1137 | 2.2K | ✔ | ✔ | | Yes/No |
| MMBench Liu et al. (2024b) | ✘ | - | 3K | ✔ | ✔ | | MC |
| OCRBench Liu et al. (2024c) | ✔ | 450 | 1K | ✔ | | | Open |
| SEED-bench-2-plus Li et al. (2024a) | ✔ | - | 2.3K | ✔ | | | MC |
| Contextual Wadhawan et al. (2024) | ✔ | 506 | 506 | ✔ | ✔ | | Open |
| MMMU iang Yue et al. (2023) | ✔ | - | 11.5K | ✔ | ✔ | | MC/Open |
| **MCTBench** | ✔ | 5.2K | 8.5K | ✔ | ✔ | ✔ | MC/Open |

Table 1: The comparison between MCTBench and previous benchmarks. **Open** and **MC** respectively present open-ended and multiple choice format for answer type. **QAs** stands for question-answer pairs. Text-Rich Oriented indicates whether the benchmark focuses on text-rich visual scenes.

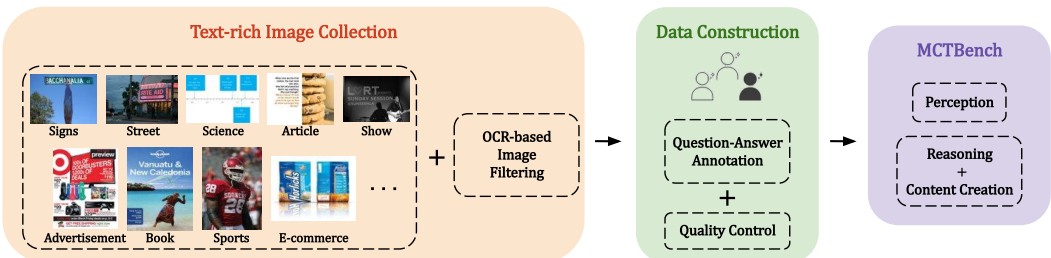

Figure 2: The pipeline of constructing MCTBench.

Zhang et al. (2024); Li et al. (2024c); Tang et al. (2024) have incorporated high-quality, text-rich visual tuning data to refine these models further.

## 2.2 MLLM BENCHMARKS

As multimodal large language models (MLLMs) continue to exhibit cross-task generality, single-task evaluations (e.g., Goyal et al. (2017); Singh et al. (2019); Chen et al. (2015); Lin et al. (2024)) are inadequate for a comprehensive performance assessment. Recent works Liu et al. (2024b); Li et al. (2023b); Fu et al. (2024); Yu et al. (2023) present general MLLM benchmarks comprising multiple tasks. Furthermore, to explore the performance of MLLMs on more complex tasks, MathVista Lu et al. (2024b) evaluates their mathematical abilities, and MMMU iang Yue et al. (2023) integrates multiple-discipline questions to benchmark MLLMs in expert domains.

Conversely, text-rich visual scenes are attracting growing attention due to their potential applications. Early works Mathew et al. (2021); Mishra et al. (2019a); Singh et al. (2019) focused on single tasks, while OCRBench Liu et al. (2024c) integrates multiple single-task datasets into five representative OCR(Optical Character Recognition)-based tasks. In contrast, our work evaluates MLLMs on complex tasks beyond OCR-based ones in text-rich visual scenes. A similar work is presented in Wadhawan et al. (2024), which demonstrates the model's performance on reasoning tasks but only on a limited set of test datasets. Our study provides a broader evaluation of cognition in text-rich visual scenes, pushing the boundaries of what MLLMs can achieve in more diverse scenarios such as content-creation. Table 1 demonstrates the detailed comparison between ours and previous benchmarks.

## 3 MCTBENCH

In this section, we outline the process of constructing the MCTBench. Section 3.1 provides an overview of MCTBench and compares it with previous benchmarks. Section 3.2 describes the procedure of collecting text-rich image sources from publicly accessible datasets. Finally, Section 3.2 explains the annotation process applied to the collected images.

## 3.1 OVERVIEW

The MCTBench is designed to evaluate the cognitive capabilities of Multimodal Large Language Models (MLLMs) towards text-rich visual scenes. To construct the comprehensive and diverse benchmark, we collected 5,194 images from a variety of public datasets, encompassing a wide array of text-rich scenes such as natural environments, books, scientific contexts, advertisements, e-commerce, and video shots. We meticulously annotate these images with a total of 8.5k question-answer pairs categorized into three tasks: perception, reasoning, and content creation. Specifically, MCTBench consists of 2,734 perception multiple-choice samples, 2,602 reasoning multiple-choice samples, and 3,130 content-creation samples. Figure 2 illustrates the overall construction pipeline of MCTBench.

## 3.2 TEXT-RICH IMAGES COLLECTION

**Image Source**    The images of MCTBench are collected from 10 different publicly available datasets, aiming to incorporate comprehensive visual scenes to evaluate the cognition of Multimodal Large Language Models (MLLMs). We begin with sampling common general natural scenes (e.g., street views, competitions, road signs) from the COCO Lin et al. (2015), Flickr30k Young et al. (2014), GQA Hudson & Manning (2019), SeedBench Li et al. (2023b) and Visual Genome Krishna et al. (2016) datasets.

Furthermore, we select conventional text-rich multimodal datasets: OCR-VQA Mishra et al. (2019b), and VizWiz Gurari et al. (2018) taken by blind photographers. To further diversify MCTBench, we incorporate three domain-specific scenes with broad potential applications: advertising (AutoUnderAds Hussain et al. (2017)), e-commerce (FoodLogoDet-1500 Hou et al. (2021)), and science (ScienceQA Lu et al. (2022)). We randomly extract one frame from each video in the AutoUnderAds dataset. All data sources are specifically selected from the testsets. We adhere to the original licenses stated by all datasets.

**OCR-based image filtering**. We select the text-rich images from the sourced images, which are guided by the following guidelines. To maintain the clarity and substantive textual content, we only retain images with valid OCR-recognized characters (with recognition probabilities higher than 0.2) of at least 10 characters. To ensure text contributes to overall visual semantics, we select images where text regions occupy more than 10% of the image area, after validating the impact of valid text lines on semantic expression. These meticulous selection criteria resulted in a curated collection of high-quality and crystal-clear text-rich images, designed to challenge and inspire advancements in perceptual and cognitive understanding within textual domains.

**Annotation**    Considering the bias and efficiency of the manual annotation, we employ a GPT-aided approach to generate at least 10 pseudo-questions for each image, and ask annotators to remove low-quality ones. All answers are human-annotated with two rounds: (1) Each image with at least 10 GPT-aided pseudo-questions, is randomly assigned to three annotators. Each annotator independently annotates the questions and provides answers. (2) Quality checkers will review the annotation in the first round. If any question or image quality does not meet our annotation guidelines, the set is re-annotated by the corresponding annotators, who also revise their answers. Annotators in the second round are required to have at least 2 years of experience in text-rich multimodal scene annotation. To reach an agreement, we use majority voting to determine the final answer. If majority voting does not reach an agreement (i.e., all three answers are inconsistent), we check if the discrepancy originated in the second round. If so, the question is re-annotated; if inconsistencies persist, it is discarded. The question is discarded if the discrepancy does not arise in the second round. In addition to content-creation tasks, due to the inherent diversity of responses, we do not provide unified answers. Instead, we offer standard references generated by powerful MLLMs (e.g., GPT-4V) and meticulously reviewed by humans.

**Quality control**    During the image quality assessment, annotators remove low-quality images (e.g., blurry, unclear text, inappropriate, solely tables/documents, and only watermark). For the QA quality assessment, annotators eliminate low-quality questions (e.g., ambiguous, overly generalized, too simplistic) and check the correctness of annotated answers (e.g., logical errors). On the other hand, We filter out multiple-choice options with more than 30% word count disparity and remove the

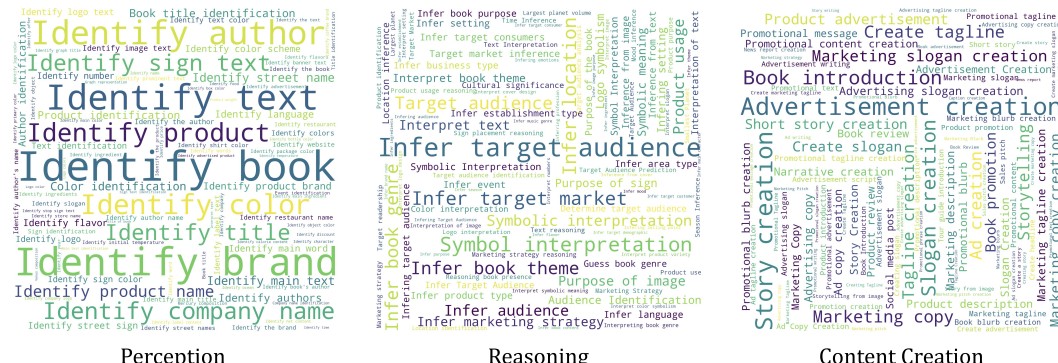

Perception       Reasoning      Content Creation

Figure 3: Visualization of the question for three different tasks using word clouds. In the word cloud, the size of a word indicates how frequently it appears. Best viewed in color.

questions that GPT-4 refuses to answer due to ethical concerns. To detail the specific issues addressed by each type, we assigned a label to each question and visualized the terms in Figure 3.

### 3.3 DATA CONSTRUCTION

## 4 EXPERIMENTS

In this section, we conduct comprehensive experiments to evaluate current MLLMs. Firstly, we outline experimental settings for the evaluated models and metrics used on MCTBench in Section 4.1 and 4.2. The results obtained from these experiments allow us to perform analysis of the selected models across three categories of tasks in Section 4.3. Furthermore, we conduct a case study to investigate performance variations among diverse models in Section 4.5.

### 4.1 MODELS

To evaluate the performance of various MLLMs on MCTBench, we select a diverse range of models categorized into two primary types: **general MLLMs** and **text-enhanced MLLMs** (specifically optimized for text recognition in images). We establish two naive baselines (random choice and frequency choice) as reference points to ensure the robustness and validity of the dataset in Table 2. Random choice involves selecting an option at random as the prediction result, while frequency choice involves selecting the prediction result based on the option with the highest proportion in the ground truth.

**General MLLMs** The experiment initially evaluates popular closed-source models, specifically Gemini-Pro Team et al. (2023) and GPT-4V(ision) OpenAI (2023). For open-source models, we select several notable general-purpose MLLMs, including Sharegpt4V Chen et al. (2023b), Honeybee Cha et al. (2024), LLaVA Liu et al. (2024a; 2023b), Otter Li et al. (2023a), Yi-VL AI et al. (2024), Qwen-VL-chat Bai et al. (2023) and Deepseek-VL Lu et al. (2024a) as competitive baselines. Additionally, we incorporate CogVLM Wang et al. (2023) and SPHINX-v2 Lin et al. (2023), which are enhanced for fine-grained understanding. To assess differentiated performance, we also integrate a larger open-source model, Mini-Gemini Li et al. (2024b).

**Text-enhanced MLLMs** Recent researchers propose remarkable works to tackle the understanding of text-rich images via enhancing the textual capabilities. Consequently, we select models mPLUG-DocOwl Ye et al. (2023b), Monkey Liu et al. (2024d); Li et al. (2024c), InternLM-XComposer2-VL Dong et al. (2024), CogAgent Hong et al. (2023) and LLaVA-NeXT Liu et al. (2024a) which have demonstrated strong OCR capabilities in previous evaluations.

## 4.2 METRICS

MCTBench is constructed with three categories of tasks: perception, reasoning, and content-creation. Due to the standard answer provided in perception and reasoning tasks, each QA pair is designed in a multiple-choice format. In contrast, the content-creation task is considered as an open-ended generation problem due to the diversity of responses.

**Perception and reasoning**   Perception and reasoning tasks entail the acquisition of information from input data, comprehension of images and text, and derivation of conclusions. Consequently, employing multiple-choice question-answering can effectively validate the corresponding capabilities of MLLMs, following Liu et al. (2024b); Li et al. (2023b). In practice, we prioritize original prompts employed by MLLMs and, if not specified, use those prompts which yield optimal results. For lengthy responses, we use regular expressions and supplementary rules to extract the option answers. We use mean accuracy to evaluate MLLMs' perception and reasoning capability.

**Content-creation**   To ensure consistency and efficiency in evaluation, we implement automatic evaluation for content-creation tasks. However, due to the diversity of answers in content creation, standard responses are not feasible, leading us to establish competitive references. These references are generated through manually crafted responses from text-only GPT-4, based on inputs from OCR recognition and detailed descriptions. The evaluation is grounded on four principal aspects: relevance, faithfulness, creativity, and instruction following. Subsequently, we employ machines (e.g., GPT-4V) to compare other MLLMs against our references with the mentioned principal, and categorize their performance as Good, Same, or Bad (i.e., the GSB metric). We measure each model's performance by calculating the percentage of 'Good' and 'Same' ratings relative to all questions, indicating how many outperform or match the constructed reference (i.e., $(G + S)/(G + S + B)$).

To assess correlations and validate the reliability of machine evaluation, we also conduct a manual evaluation on subsets of the data using the mentioned metrics (GSB) and compare them with machine evaluation. We integrate three powerful MLLMs (GPT-4V OpenAI (2023), Gemini-Pro Team et al. (2023) and LLaVA-NeXT Liu et al. (2024a)) as evaluators. We measure the evaluation correlation between humans and machines using accuracy and Pearson correlation Benesty et al. (2009) on GSB scores. Table 3 illustrates the results between three machine evaluators and human evaluations, indicating that GPT-4V achieves a top correlation score.

## 4.3 RESULTS

**Perception**   Firstly, experiments are conducted to verify the perceptual performances of each model as baselines. As shown in Table 2, most models achieved satisfying scores on the perception task. Closed-source models (e.g., GPT-4V OpenAI (2023)) demonstrated excellent accuracy, while some open-source models (e.g., Mini-Gemini Li et al. (2024b)) demonstrated superior perception capabilities, surpassing the their performance. Models with higher resolutions and more parameters (e.g., LLaVA-NeXT Liu et al. (2024a) and Mini-Gemini Li et al. (2024b)), typically performed better. Among similarly-sized models, text-enhanced MLLMs generally outperformed others by effectively extracting text from images and generating precise responses.

**Reasoning**   The reasoning task is more challenging than perception. It requires not only effective extraction and fusion of visual and textual features, but also involves comprehensive inference to generate accurate responses. As shown in Table 2, most models exhibited a significant drop in scores due to the increased difficulty of reasoning tasks compared to perception tasks.

Notably, GPT-4V demonstrates exceptional performance among MLLMs, surpassing most models by a significant margin. Besides, there still exists a positive correlation between a model's performance and the number of its parameters. This phenomenon arises from larger models' enhanced ability, to comprehend text and integrate image information for reasoning more effectively.

Nevertheless, text-enhanced MLLMs have not substantially outperformed general models on reasoning tasks. Given that text-enhanced models have achieved superior results in perception tasks, we posit their effectiveness in recognizing text within images. However, achieving higher scores in reasoning tasks necessitates the ability to analyse and summarise effectively. TextMonkey Liu et al. (2024d) shows the least performance gap and achieves results comparable to the perception

| Model | Params | Perception | Cognition | | Average Scores | | |
| --- | --- | --- | --- | --- | --- | --- | --- |
| | | | Reasoning | Content-Creation* | MC | Cog | All |
| Naive Baseline | | | | | | | |
| Random choice | - | 25.00 | 25.00 | - | - | - | - |
| Frequency choice | - | 25.16 | 25.52 | - | - | - | - |
| General MLLMs | | | | | | | |
| GPT-4V OpenAI (2023) | - | 83.58 | **74.21** | **87.35** | **78.90** | **83.12** | **81.71** |
| Gemini-Pro Team et al. (2023) | - | 78.79 | 70.18 | 56.78 | 74.49 | 65.63 | 68.58 |
| Yi-VL AI et al. (2024) | 6B | 77.25 | 72.33 | 41.45 | 74.79 | 58.12 | 63.68 |
| Deepseek-VL Lu et al. (2024a) | 7B | 76.74 | 68.79 | 57.25 | 72.77 | 65.01 | 67.59 |
| Honeybee Cha et al. (2024) | 7B | 72.60 | 67.22 | 73.64 | 69.91 | 71.78 | 71.15 |
| Otter Li et al. (2023a) | 7B | 58.12 | 54.42 | 31.70 | 56.27 | 43.99 | 48.08 |
| Qwen-VL-chat Bai et al. (2023) | 7B | 77.98 | 70.68 | 67.53 | 74.33 | 70.93 | 72.06 |
| Sharegpt4V Chen et al. (2023b) | 13B | 74.54 | 69.49 | 66.19 | 72.02 | 69.10 | 70.07 |
| LLaVA-1.5 Liu et al. (2023b) | 13B | 78.09 | 72.56 | 66.47 | 75.33 | 70.90 | 72.37 |
| SPHINX-v2 Lin et al. (2023) | 13B | 78.02 | 71.94 | 62.30 | 74.98 | 68.64 | 70.75 |
| CogVLM Wang et al. (2023) | 17B | 71.40 | 69.52 | 65.61 | 70.46 | 68.04 | 68.84 |
| Mini-Gemini Li et al. (2024b) | 34B | **83.83** | 73.33 | 86.76 | 78.58 | 82.67 | 81.31 |
| Text-enhanced MLLMs | | | | | | | |
| IXC 2 Dong et al. (2024) | 7B | 78.05 | 72.10 | 74.45 | 75.08 | 74.76 | 74.87 |
| Monkey Li et al. (2024c) | 7B | 79.22 | **72.64** | 59.56 | 75.93 | 67.75 | 70.47 |
| TextMonkey Liu et al. (2024d) | 7B | 71.80 | 69.45 | 22.81 | 70.63 | 46.72 | 54.69 |
| mPLUG-DocOwl Ye et al. (2023a) | 10B | 75.05 | 70.06 | 60.87 | 72.56 | 66.71 | 68.66 |
| CogAgent Hong et al. (2023) | 34B | 58.56 | 56.46 | 56.86 | 57.51 | 57.19 | 57.29 |
| LLaVA-NeXT Liu et al. (2024a) | 34B | **83.87** | 71.64 | **85.30** | 77.76 | 81.53 | 80.27 |

Table 2: Evaluation results for MLLMs on MCTBench. **MC** means the average score of the two tasks (perception and reasoning) in multiple-choice format. **Cog** means the average scores of the two cognitive tasks (reasoning and content-creation). **All** means the overall average scores of all tasks. *The content-creation task is scored using the percentage of 'Good' and 'Same' ratings by the GSB metric except for accuracy used in Perception and reasoning tasks. Numbers in **Bold** and underline represent the top-2 results in each task.

task. However, most text-enhanced models are not explicitly trained in this aspect, and consequently do not outperform general MLLMs.

| | GPT-4V | Gemini-Pro | Best open-source (LLaVA-NeXT) |
| --- | --- | --- | --- |
| Accuracy | 79.38 | 70.71 | 65.22 |
| Pearson Correlation | 0.558 | 0.380 | 0.304 |

Table 3: Correlation analysis between automatic machine and human evaluation on content-creation using the accuracy and Pearson correlation coefficient.

**Content-creation** The open-ended creation task differs from the aforementioned tasks, leading to the observation of a broader array of perspectives. As mentioned in Section 4.2, the evaluation is based on four principal aspects: relevance, faithfulness, creativity, and instruction following. Therefore, as the content-creation task emphasizes the generation of suitable text for images, we also notice significantly enhancement on performances by employing larger language models. However, the performance of text-enhanced MLLMs exhibits considerable variation due to differences in training objectives. Some models are specifically trained to extract structured text information or comprehend lengthy text inputs. Consequently, they may underperform relative to general models when tasked with creative endeavours. We also found that the closed-source model Gemini-Pro Team et al. (2023) is unable to achieve good results. A notable discrepancy in its scores of template following demonstrates that Gemini struggles to create corresponding formats for task specifications. For instance, in the task of generating a slogan, Gemini may tend to produce lengthy paragraphs

| Input | Perception | Reasoning | Creation |
|---|---|---|---|
| Image + OCR texts | 75.60 | 71.68 | 65.45 |
| Image (text regions removed) | 62.22 | 62.22 | 46.89 |
| Image (text regions only) | 74.69 | 68.01 | 58.78 |
| Image (baseline) | 78.09 | 72.56 | 66.47 |

Table 4: The effect of visual and textual information.

instead of concise phrases. Some text-enhanced MLLMs, such as TextMonkey, also have similar phenomena.

**Summary**    MLLMs have shown certain multi-modal capabilities in perception tasks, with mainstream models achieving commendable performances. However, in reasoning tasks, existing models still have room for improvement. Performance on content-creation tasks indicates that text-enhanced MLLMs trained for different types of tasks may lose some creative capabilities. We suggest that reasoning and creation tasks serve as distinct dimensions for evaluation, offering insights into the model's comprehension of input and its proficiency in generating output responses. While existing models excel in basic perception tasks, achieving comparable competence in reasoning and creation tasks remains challenging.

## 4.4 ABLATION STUDY

**The effect of visual and textual information.**    In Table 4, we conduct several experiments to reveal the effect of visual and textual information in images of our benchmark. We use the LLaVA-1.5 as the baseline. Firstly, we conduct an experiment that adds OCR texts as input. As shown in the table, even if the OCR texts are contained as input, the model cannot gain explicit improvement. This demonstrates that the MCTBench does not rely solely on texts to get answers. Furthermore, we conduct an experiment that removes all texts from images to check the importance of texts in our benchmark. Specifically, we detected and blurred all the texts in images. We noticed that the performance dropped significantly. This also demonstrates that it is difficult for a model to correctly answer the question without text in the image. Additionally, if we only keep the text region, and delete other background parts in the image. The performance is between the above two experiments, indicating that MCTBench relies both on textual and visual information, to get the final result. To conclude, explicitly adding OCR texts, or removing text/image parts does not help, or even lead to worse performance on MCTBench. MLLMs are required to jointly recognize related textual and visual patterns, to answer the questions correctly.

## 4.5 CASE STUDY

For the perception and reasoning tasks, we select representative cases shown in Figure 4. Specifically, we select GPT-4V as a strong reference along with several open-source representative MLLMs, split into three groups: large model size and resolution MLLMs (Mini-Gemini Li et al. (2024b) and LLaVA-NeXT Liu et al. (2024a)), text-enhanced MLLMs (Monkey Li et al. (2024c) and mPLUG-DocOwl Ye et al. (2023a)), and general MLLMs (LLaVA-1.5 Liu et al. (2023a) and ShareGPT4V Chen et al. (2023b)). For the selected perception question, both high-resolution MLLMs and text-enhanced MLLMs perform well, while general MLLMs fail in fine-grained understanding. On the contrary, text-enhanced MLLMs excelled in perception but performed poorly in reasoning tasks. Larger models like Mini-Gemini Li et al. (2024b), LLaVA-NeXT Liu et al. (2024a), and GPT-4V OpenAI (2023) handle reasoning better by effectively integrating textual and visual elements.

For the content-creation task, using GPT-4V OpenAI (2023) as a robust reference, we select general and text-enhanced models that performed well on MCTBench, and conduct case studies in three scenarios. As shown in Figure 5, GPT-4V OpenAI (2023) significantly surpassed other models in content creation quality. Mini-Gemini Li et al. (2024b) also showed consistent performance across cognitive tasks, while text-enhanced models like Monkey Li et al. (2024c) were limited to text recognition and basic descriptions, resulting in less attractive content.

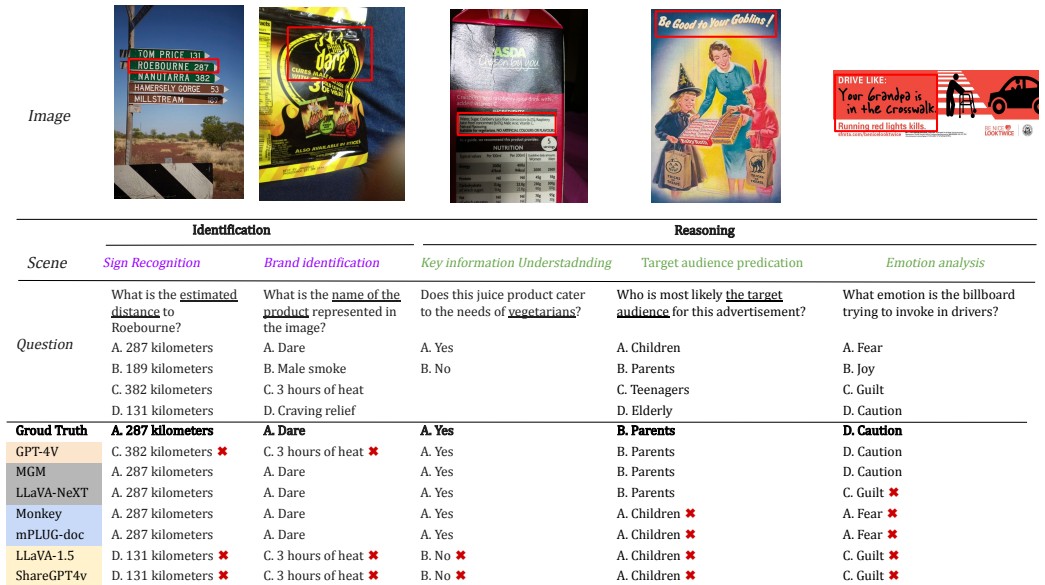

| | Identification | | Reasoning | | |
|---|---|---|---|---|---|
| Scene | *Sign Recognition* | *Brand identification* | *Key information Understadnding* | *Target audience predication* | *Emotion analysis* |
| Question | What is the estimated distance to Roebourne? | What is the name of the product represented in the image? | Does this juice product cater to the needs of vegetarians? | Who is most likely the target audience for this advertisement? | What emotion is the billboard trying to invoke in drivers? |
| | A. 287 kilometers | A. Dare | A. Yes | A. Children | A. Fear |
| | B. 189 kilometers | B. Male smoke | B. No | B. Parents | B. Joy |
| | C. 382 kilometers | C. 3 hours of heat | | C. Teenagers | C. Guilt |
| | D. 131 kilometers | D. Craving relief | | D. Elderly | D. Caution |
| **Groud Truth** | **A. 287 kilometers** | **A. Dare** | **A. Yes** | **B. Parents** | **D. Caution** |
| GPT-4V | C. 382 kilometers ✖ | C. 3 hours of heat ✖ | A. Yes | B. Parents | D. Caution |
| MGM | A. 287 kilometers | A. Dare | A. Yes | B. Parents | D. Caution |
| LLaVA-NeXT | A. 287 kilometers | A. Dare | A. Yes | B. Parents | C. Guilt ✖ |
| Monkey | A. 287 kilometers | A. Dare | A. Yes | A. Children ✖ | A. Fear ✖ |
| mPLUG-doc | A. 287 kilometers | A. Dare | A. Yes | A. Children ✖ | A. Fear ✖ |
| LLaVA-1.5 | D. 131 kilometers ✖ | C. 3 hours of heat ✖ | B. No ✖ | A. Children ✖ | C. Guilt ✖ |
| ShareGPT4v | D. 131 kilometers ✖ | C. 3 hours of heat ✖ | B. No ✖ | A. Children ✖ | C. Guilt ✖ |

Figure 4: The cases of predication from different MLLMs divided into four groups: GPT-4V OpenAI (2023), Mini-Gemini Li et al. (2024b)(MGM) and LLaVA-NeXT Liu et al. (2024a) for larger model size, Monkey Li et al. (2024c) and mPLUG-DocOwl Ye et al. (2023a) for text-enhanced MLLMs, LLaVA-1.5 Liu et al. (2023a) and ShareGPT4V Chen et al. (2023b) for the general MLLMs

## 5 LIMITATIONS

Our dataset primarily focuses on English, which may limit the generalization of our findings to multilingual scenes. Although we believe that the cognitive capacities of MLLMs should theoretically extend to other languages, we have not empirically substantiated this assertion in the present study. Additionally, we have only selected a subset of representative models for evaluation due to space constraints. This selection may not cover the full spectrum of currently available MLLMs. Our future work aims to provide evaluation results for a more extensive range of models.

## 6 CONCLUSION

In this work, we introduce MCTBench, a comprehensive benchmark designed to evaluate the cognitive capabilities of MLLMs in text-rich visual scenes. The MCTBench comprises 5.2k images and 8.5k question-answer pairs, covering a range of tasks including reasoning, content creation for cognitive assessment, and conventional perception. Evaluations of MLLMs on MCTBench reveal that current MLLMs still need further advancements in cognitive capabilities, despite their superior perception performance. We hope that MCTBench will motivate researchers to further improve the cognitive capabilities of MLLMs in text-rich visual scenes, thereby enhancing the practical utility of AI in real-world applications.

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

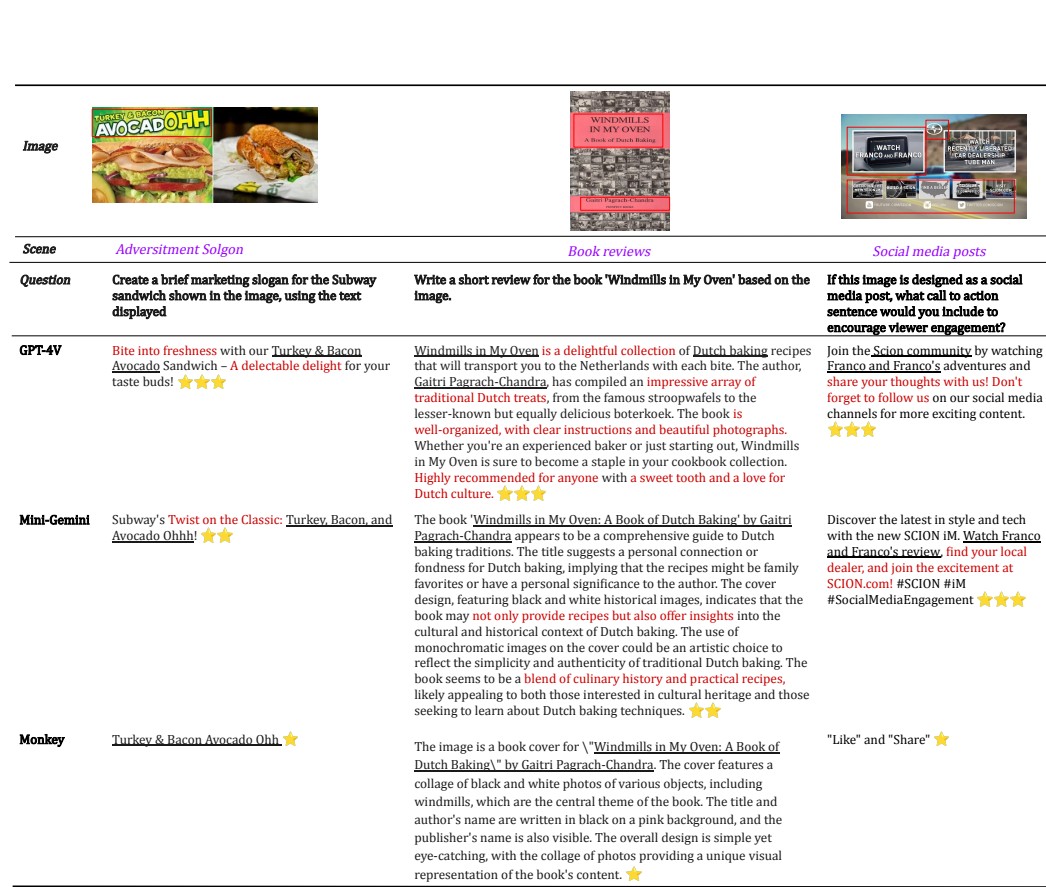

Figure 5: The cases of predication on content-creation tasks from three representative MLLMs: GPT-4V OpenAI (2023), Mini-Gemini Li et al. (2024b) and Monkey Li et al. (2024c). We mark high-quality sentences in red, words hit the text in the image with underlining, and rate the quality of the generation with stars.

Cabi, Tengda Han, Zhitao Gong, Sina Samangooei, Marianne Monteiro, Jacob Menick, Sebastian Borgeaud, Andrew Brock, Aida Nematzadeh, Sahand Sharifzadeh, Mikolaj Binkowski, Ricardo Barreira, Oriol Vinyals, Andrew Zisserman, and Karen Simonyan. Flamingo: a visual language model for few-shot learning, 2022.

Jinze Bai, Shuai Bai, Shusheng Yang, Shijie Wang, Sinan Tan, Peng Wang, Junyang Lin, Chang Zhou, and Jingren Zhou. Qwen-vl: A frontier large vision-language model with versatile abilities. *arXiv preprint arXiv:2308.12966*, 2023.

Jacob Benesty, Jingdong Chen, Yiteng Huang, and Israel Cohen. Pearson correlation coefficient. In *Noise reduction in speech processing*, pp. 37–40. Springer, 2009.

Junbum Cha, Wooyoung Kang, Jonghwan Mun, and Byungseok Roh. Honeybee: Locality-enhanced projector for multimodal llm. In *Proceedings of the IEEE/CVF Conference on Computer Vision and Pattern Recognition (CVPR)*, 2024.

Jun Chen, Deyao Zhu, Xiaoqian Shen, Xiang Li, Zechun Liu, Pengchuan Zhang, Raghuraman Krishnamoorthi, Vikas Chandra, Yunyang Xiong, and Mohamed Elhoseiny. Minigpt-v2: large language model as a unified interface for vision-language multi-task learning. *arXiv preprint arXiv:2310.09478*, 2023a.

Lin Chen, Jisong Li, Xiaoyi Dong, Pan Zhang, Conghui He, Jiaqi Wang, Feng Zhao, and Dahua Lin. Sharegpt4v: Improving large multi-modal models with better captions. *arXiv preprint arXiv:2311.12793*, 2023b.

Xinlei Chen, Hao Fang, Tsung-Yi Lin, Ramakrishna Vedantam, Saurabh Gupta, Piotr Dollar, and C. Lawrence Zitnick. Microsoft coco captions: Data collection and evaluation server, 2015.

Wei-Lin Chiang, Zhuohan Li, Zi Lin, Ying Sheng, Zhanghao Wu, Hao Zhang, Lianmin Zheng, Siyuan Zhuang, Yonghao Zhuang, Joseph E. Gonzalez, Ion Stoica, and Eric P. Xing. Vicuna: An open-source chatbot impressing gpt-4 with 90%* chatgpt quality, March 2023. URL https://lmsys.org/blog/2023-03-30-vicuna/.

Wenliang Dai, Junnan Li, Dongxu Li, Anthony Meng Huat Tiong, Junqi Zhao, Weisheng Wang, Boyang Li, Pascale Fung, and Steven Hoi. Instructblip: Towards general-purpose vision-language models with instruction tuning, 2023.

Xiaoyi Dong, Pan Zhang, Yuhang Zang, Yuhang Cao, Bin Wang, Linke Ouyang, Xilin Wei, Songyang Zhang, Haodong Duan, Maosong Cao, Wenwei Zhang, Yining Li, Hang Yan, Yang Gao, Xinyue Zhang, Wei Li, Jingwen Li, Kai Chen, Conghui He, Xingcheng Zhang, Yu Qiao, Dahua Lin, and Jiaqi Wang. Internlm-xcomposer2: Mastering free-form text-image composition and comprehension in vision-language large model. *arXiv preprint arXiv:2401.16420*, 2024.

Hao Feng, Zijian Wang, Jingqun Tang, Jinghui Lu, Wengang Zhou, Houqiang Li, and Can Huang. Unidoc: A universal large multimodal model for simultaneous text detection, recognition, spotting and understanding, 2023.

Chaoyou Fu, Peixian Chen, Yunhang Shen, Yulei Qin, Mengdan Zhang, Xu Lin, Jinrui Yang, Xiawu Zheng, Ke Li, Xing Sun, Yunsheng Wu, and Rongrong Ji. Mme: A comprehensive evaluation benchmark for multimodal large language models, 2024.

Yash Goyal, Tejas Khot, Douglas Summers-Stay, Dhruv Batra, and Devi Parikh. Making the V in VQA matter: Elevating the role of image understanding in Visual Question Answering. In *Conference on Computer Vision and Pattern Recognition (CVPR)*, 2017.

Danna Gurari, Qing Li, Abigale J. Stangl, Anhong Guo, Chi Lin, Kristen Grauman, Jiebo Luo, and Jeffrey P. Bigham. Vizwiz grand challenge: Answering visual questions from blind people, 2018.

Wenyi Hong, Weihan Wang, Qingsong Lv, Jiazheng Xu, Wenmeng Yu, Junhui Ji, Yan Wang, Zihan Wang, Yuxiao Dong, Ming Ding, et al. Cogagent: A visual language model for gui agents. *arXiv preprint arXiv:2312.08914*, 2023.

Qiang Hou, Weiqing Min, Jing Wang, Sujuan Hou, Yuanjie Zheng, and Shuqiang Jiang. Foodlogodet-1500: A dataset for large-scale food logo detection via multi-scale feature decoupling network. In *Proceedings of the 29th ACM International Conference on Multimedia*, MM '21. ACM, October 2021. doi: 10.1145/3474085.3475289. URL `http://dx.doi.org/10.1145/3474085.3475289`.

Wenbo Hu, Y. Xu, Y. Li, W. Li, Z. Chen, and Z. Tu. Bliva: A simple multimodal llm for better handling of text-rich visual questions. In *AAAI Conference on Artificial Intelligence*, 2023. URL `https://api.semanticscholar.org/CorpusID:261049015`.

Drew A. Hudson and Christopher D. Manning. Gqa: A new dataset for real-world visual reasoning and compositional question answering, 2019.

Zaeem Hussain, Mingda Zhang, Xiaozhong Zhang, Keren Ye, Christopher Thomas, Zuha Agha, Nathan Ong, and Adriana Kovashka. Automatic understanding of image and video advertisements. *2017 IEEE Conference on Computer Vision and Pattern Recognition (CVPR)*, pp. 1100–1110, 2017. URL `https://api.semanticscholar.org/CorpusID:11172071`.

iang Yue, Yuansheng Ni, Kai Zhang, Tianyu Zheng, Ruoqi Liu, Ge Zhang, Samuel Stevens, Dongfu Jiang, Weiming Ren, Yuxuan Sun, Cong Wei, Botao Yu, Ruibin Yuan, Renliang Sun, Ming Yin, Boyuan Zheng, Zhenzhu Yang, Yibo Liu, Wenhao Huang, Huan Sun, Yu Su, and Wenhu Chen. Mmmu: A massive multi-discipline multimodal understanding and reasoning benchmark for expert agi, 2023.

Ranjay Krishna, Yuke Zhu, Oliver Groth, Justin Johnson, Kenji Hata, Joshua Kravitz, Stephanie Chen, Yannis Kalantidis, Li-Jia Li, David A. Shamma, Michael S. Bernstein, and Fei-Fei Li. Visual genome: Connecting language and vision using crowdsourced dense image annotations, 2016.

Bo Li, Yuanhan Zhang, Liangyu Chen, Jinghao Wang, Jingkang Yang, and Ziwei Liu. Otter: A multi-modal model with in-context instruction tuning. *ArXiv*, abs/2305.03726, 2023a. URL `https://api.semanticscholar.org/CorpusID:258547300`.

Bohao Li, Rui Wang, Guangzhi Wang, Yuying Ge, Yixiao Ge, and Ying Shan. Seed-bench: Benchmarking multimodal llms with generative comprehension, 2023b.

Bohao Li, Yuying Ge, Yi Chen, Yixiao Ge, Ruimao Zhang, and Ying Shan. Seed-bench-2-plus: Benchmarking multimodal large language models with text-rich visual comprehension, 2024a.

Junnan Li, Dongxu Li, Silvio Savarese, and Steven Hoi. Blip-2: Bootstrapping language-image pre-training with frozen image encoders and large language models, 2023c.

Yanwei Li, Yuechen Zhang, Chengyao Wang, Zhisheng Zhong, Yixin Chen, Ruihang Chu, Shaoteng Liu, and Jiaya Jia. Mini-gemini: Mining the potential of multi-modality vision language models, 2024b.

Zhang Li, Biao Yang, Qiang Liu, Zhiyin Ma, Shuo Zhang, Jingxu Yang, Yabo Sun, Yuliang Liu, and Xiang Bai. Monkey: Image resolution and text label are important things for large multi-modal models. In *proceedings of the IEEE/CVF conference on computer vision and pattern recognition*, 2024c.

Kun-Yu Lin, Henghui Ding, Jiaming Zhou, Yi-Xing Peng, Zhilin Zhao, Chen Change Loy, and Wei-Shi Zheng. Rethinking clip-based video learners in cross-domain open-vocabulary action recognition. *arXiv preprint arXiv:2403.01560*, 2024.

Tsung-Yi Lin, Michael Maire, Serge Belongie, Lubomir Bourdev, Ross Girshick, James Hays, Pietro Perona, Deva Ramanan, C. Lawrence Zitnick, and Piotr Dollár. Microsoft coco: Common objects in context, 2015.

Ziyi Lin, Chris Liu, Renrui Zhang, Peng Gao, Longtian Qiu, Han Xiao, Han Qiu, Chen Lin, Wenqi Shao, Keqin Chen, Jiaming Han, Siyuan Huang, Yichi Zhang, Xuming He, Hongsheng Li, and Yu Qiao. Sphinx: The joint mixing of weights, tasks, and visual embeddings for multi-modal large language models, 2023.

Haotian Liu, Chunyuan Li, Yuheng Li, and Yong Jae Lee. Improved baselines with visual instruction tuning, 2023a.

Haotian Liu, Chunyuan Li, Qingyang Wu, and Yong Jae Lee. Visual instruction tuning, 2023b.

Haotian Liu, Chunyuan Li, Yuheng Li, Bo Li, Yuanhan Zhang, Sheng Shen, and Yong Jae Lee. Llava-next: Improved reasoning, ocr, and world knowledge, January 2024a. URL https://llava-vl.github.io/blog/2024-01-30-llava-next/.

Yuan Liu, Haodong Duan, Yuanhan Zhang, Bo Li, Songyang Zhang, Wangbo Zhao, Yike Yuan, Jiaqi Wang, Conghui He, Ziwei Liu, Kai Chen, and Dahua Lin. Mmbench: Is your multi-modal model an all-around player?, 2024b.

Yuliang Liu, Zhang Li, Biao Yang, Chunyuan Li, Xucheng Yin, Cheng lin Liu, Lianwen Jin, and Xiang Bai. On the hidden mystery of ocr in large multimodal models, 2024c.

Yuliang Liu, Biao Yang, Qiang Liu, Zhang Li, Zhiyin Ma, Shuo Zhang, and Xiang Bai. Textmonkey: An ocr-free large multimodal model for understanding document. *arXiv preprint arXiv:2403.04473*, 2024d.

Haoyu Lu, Wen Liu, Bo Zhang, Bingxuan Wang, Kai Dong, Bo Liu, Jingxiang Sun, Tongzheng Ren, Zhuoshu Li, Hao Yang, Yaofeng Sun, Chengqi Deng, Hanwei Xu, Zhenda Xie, and Chong Ruan. Deepseek-vl: Towards real-world vision-language understanding, 2024a.

Pan Lu, Swaroop Mishra, Tony Xia, Liang Qiu, Kai-Wei Chang, Song-Chun Zhu, Oyvind Tafjord, Peter Clark, and Ashwin Kalyan. Learn to explain: Multimodal reasoning via thought chains for science question answering. In *The 36th Conference on Neural Information Processing Systems (NeurIPS)*, 2022.

Pan Lu, Hritik Bansal, Tony Xia, Jiacheng Liu, Chunyuan Li, Hannaneh Hajishirzi, Hao Cheng, Kai-Wei Chang, Michel Galley, and Jianfeng Gao. Mathvista: Evaluating mathematical reasoning of foundation models in visual contexts, 2024b.

Minesh Mathew, Dimosthenis Karatzas, and CV Jawahar. Docvqa: A dataset for vqa on document images. In *Proceedings of the IEEE/CVF winter conference on applications of computer vision*, pp. 2200–2209, 2021.

Brandon McKinzie, Zhe Gan, Jean-Philippe Fauconnier, Sam Dodge, Bowen Zhang, Philipp Dufter, Dhruti Shah, Xianzhi Du, Futang Peng, Floris Weers, et al. Mm1: Methods, analysis & insights from multimodal llm pre-training. *arXiv preprint arXiv:2403.09611*, 2024.

Anand Mishra, Shashank Shekhar, Ajeet Kumar Singh, and Anirban Chakraborty. Ocr-vqa: Visual question answering by reading text in images. In *2019 international conference on document analysis and recognition (ICDAR)*, pp. 947–952. IEEE, 2019a.

Anand Mishra, Shashank Shekhar, Ajeet Kumar Singh, and Anirban Chakraborty. Ocr-vqa: Visual question answering by reading text in images. In *ICDAR*, 2019b.

OpenAI. Gpt-4 technical report. 2023. URL https://api.semanticscholar.org/CorpusID:257532815.

Amanpreet Singh, Vivek Natarajan, Meet Shah, Yu Jiang, Xinlei Chen, Dhruv Batra, Devi Parikh, and Marcus Rohrbach. Towards vqa models that can read. In *Proceedings of the IEEE/CVF Conference on Computer Vision and Pattern Recognition (CVPR)*, June 2019.

Jingqun Tang, Chunhui Lin, Zhen Zhao, Shu Wei, Binghong Wu, Qi Liu, Hao Feng, Yang Li, Siqi Wang, Lei Liao, et al. Textsquare: Scaling up text-centric visual instruction tuning. *arXiv preprint arXiv:2404.12803*, 2024.

Gemini Team, Rohan Anil, Sebastian Borgeaud, Yonghui Wu, Jean-Baptiste Alayrac, Jiahui Yu, Radu Soricut, Johan Schalkwyk, Andrew M Dai, Anja Hauth, et al. Gemini: a family of highly capable multimodal models. *arXiv preprint arXiv:2312.11805*, 2023.

Hugo Touvron, Thibaut Lavril, Gautier Izacard, Xavier Martinet, Marie-Anne Lachaux, Timothée Lacroix, Baptiste Rozière, Naman Goyal, Eric Hambro, Faisal Azhar, Aurelien Rodriguez, Armand Joulin, Edouard Grave, and Guillaume Lample. Llama: Open and efficient foundation language models. *ArXiv*, abs/2302.13971, 2023. URL `https://api.semanticscholar.org/CorpusID:257219404`.

Rohan Wadhawan, Hritik Bansal, Kai-Wei Chang, and Nanyun Peng. Contextual: Evaluating context-sensitive text-rich visual reasoning in large multimodal models, 2024.

Weihan Wang, Qingsong Lv, Wenmeng Yu, Wenyi Hong, Ji Qi, Yan Wang, Junhui Ji, Zhuoyi Yang, Lei Zhao, Xixuan Song, et al. Cogvlm: Visual expert for pretrained language models. *arXiv preprint arXiv:2311.03079*, 2023.

Haoran Wei, Lingyu Kong, Jinyue Chen, Liang Zhao, Zheng Ge, Jinrong Yang, Jianjian Sun, Chunrui Han, and Xiangyu Zhang. Vary: Scaling up the vision vocabulary for large vision-language models, 2023.

Jiabo Ye, Anwen Hu, Haiyang Xu, Qinghao Ye, Ming Yan, Yuhao Dan, Chenlin Zhao, Guohai Xu, Chenliang Li, Junfeng Tian, Qian Qi, Ji Zhang, and Fei Huang. mplug-docowl: Modularized multimodal large language model for document understanding, 2023a.

Qinghao Ye, Haiyang Xu, Jiabo Ye, Ming Yan, Anwen Hu, Haowei Liu, Qi Qian, Ji Zhang, Fei Huang, and Jingren Zhou. mplug-owl2: Revolutionizing multi-modal large language model with modality collaboration, 2023b.

Peter Young, Alice Lai, Micah Hodosh, and Julia Hockenmaier. From image descriptions to visual denotations: New similarity metrics for semantic inference over event descriptions. *TACL*, 2:67–78, 2014.

Weihao Yu, Zhengyuan Yang, Linjie Li, Jianfeng Wang, Kevin Lin, Zicheng Liu, Xinchao Wang, and Lijuan Wang. Mm-vet: Evaluating large multimodal models for integrated capabilities, 2023.

Yanzhe Zhang, Ruiyi Zhang, Jiuxiang Gu, Yufan Zhou, Nedim Lipka, Diyi Yang, and Tong Sun. Llavar: Enhanced visual instruction tuning for text-rich image understanding, 2024.

