# OpenReview forum: "MCTBench: Multimodal Cognition towards Text-Rich Visual Scenes Benchmark"
_ICLR.cc/2025/Conference — Submitted to ICLR 2025_

### Official Review · Reviewer_18nb · 2024-10-22

**Soundness:** 3
**Presentation:** 1
**Contribution:** 3
**Rating:** 3
**Confidence:** 4

**Summary:**

The paper introduces MCTBench, a novel benchmark, with 8.5k QA pairs, aiming at evaluating the cognitive abilities of VLMs in text-rich visual scenes through visual reasoning and content creation tasks. Using GPT-4V to assist annotators in improving data quality and evaluate content creation efficiently. Several experiments on 18 VLMs are provided and pointing out that text-enhanced VLMs trained for different types of tasks may lose some creative capabilities of content creation.

**Strengths:**

1. MCTBench fills an important gap by introducing content creation task to evaluate cognitive capabilities of VLMs in text-rich scenes.
2. Authors provide a large, diverse and high-quality human annotated dataset containing perception, reasoning, and content creation tasks.
3. The testing results on 18 VLMs inform readers that there is still room for improvement in reasoning tasks for current VLMs and in content creation tasks, text-enhanced VLMs trained for different types of tasks may lose some creative capabilities.

**Weaknesses:**

1. More newer VLMs, such as Gemini 1.5 Pro (Feb. 2024), InternVL1.5-Chat (Apr. 2024), GPT-4o (May 2024) and Claude 3.5 Sonnet (Jun. 2024) should be considered.
2. The reliability of automated evaluation using GPT-4V is questioned.
3. The paper lacks further insightful analyses, such as the impact of the resolution of source images on the results, the impact of different language decoders on the results of content creation task.

**Questions:**

1. In Table 2, the performance difference between strong models, like GPT-4V and weaker models, like LLaVA1.5-13B is minimal on reasoning tasks. What could be the cause of this result?
2. Is the 79.38 accuracy for GPT-4V to evaluate on content creation task higher enough to replace humans? Could you provide the accuracy of human evaluation?
3. Authors are encouraged to provide results on some of the latest models, such as the InternVL2 series (2024/07/04), and the Qwen2-VL series (2024/08/30). While the results of these models are not mandatory under the guidelines, considering the super-fast advancements in VLMs this year. Could you please include results from some of the aforementioned models to highlight the performance of the latest generation of VLMs?
4. Refer to Weaknesses 3.
5. There is no content in Section 3.3 Data Construction.
6. Figure 5 is in Reference. Authors are encouraged to reformat it.

---

### Official Review · Reviewer_WzK4 · 2024-10-28

**Soundness:** 2
**Presentation:** 1
**Contribution:** 3
**Rating:** 3
**Confidence:** 4

**Summary:**

This paper present MCTBench, a new multimodal benchmark designed to evaluate the cognitive abilities of MLLMs through visual reasoning and content creation tasks. MCTBench includes perception tasks and employs an automated evaluation process for content creation, revealing that while MLLMs exhibit strong perceptual skills, their cognitive abilities need improvement. This benchmark aims to provide a valuable tool for the community to advance cognitive capabilities in processing text-rich visual scenes.

**Strengths:**

1. This paper broadens the scope of OCR ability of MLLMs, rather than  conventional OCR tasks and current MLLM benchmarks.
2. The benchmark is large-scale and human-annotated, make the benchmark valid and reliable.

**Weaknesses:**

1. Paper is poorly formatted.
2. Paper lacks details.
See questions below.

**Questions:**

1. The paper is poorly written, with many citation format errors. Section 3.3 is incomplete, and on page ten, there is a figure inserted in the middle of the references section. Additionally, there is no appendix provided.
2. Many details are not clearly explained. For example, the content-creation task lacks sufficient explanation, and the prompts used are not detailed.
3. While the paper claims to "provide a broader evaluation of cognition in text-rich visual scenes," this is only reflected in the word cloud, lacking other relevant support. For instance, under the reasoning task, it is unclear how OCR capabilities are subdivided into fine-grained categories, nor is there a comparison with other benchmarks at a fine-grained level. It also remains unclear which aspects are covered by existing benchmarks and which are not, and if there are any additional examples for OCR abilities not covered by current benchmarks.

---

### Official Review · Reviewer_Lfto · 2024-10-29

**Soundness:** 2
**Presentation:** 2
**Contribution:** 2
**Rating:** 3
**Confidence:** 3

**Summary:**

This paper introduced MCTBench, a comprehensive benchmark designed to evaluate the cognitive capabilities of MLLMs in text-rich visual scenes. The MCTBench comprises 5.2k images and 8.5k question-answer pairs, covering a range of tasks including reasoning, content creation for cognitive assessment, and conventional perception.

**Strengths:**

1. This paper collect a large-scale benchmark for evaluating the cognitive capability for MLLM, where reasoning and content-creation ability is highlighted
2. For the content-creation task, an automated evaluation pipeline is introduced to enhance efficiency.

**Weaknesses:**

1. The paper introduces Content Creation as a new evaluation component, but it could benefit from a clearer explanation of the necessity and value of this addition for assessing cognitive abilities. Furthermore, the rationale behind dividing cognitive tasks into “reasoning” and “content creation” would be strengthened with additional justification for this categorization.
2. The paper suggests that MLLMs require improvements in cognitive capabilities within text-rich visual scenes. However, the results presented do not entirely support this conclusion, as cognitive scores do not show a substantial decrease compared to perceptual scores. Since cognition often builds on perception, the separation of these tasks across different data samples may seem too rigid. Evaluating perception and cognition on the same images could better capture their relationship and provide clearer insights into how MLLMs leverage perceptual understanding for reasoning.
3. The automatic evaluation approach could be better supported by a further improvement and a comparison to prior evaluation methods. Specifically, with a Pearson correlation of only 0.558 against human judgment, this score may be insufficient to fully validate the reliability of the automated approach. A higher correlation score would likely provide stronger validation.

**Questions:**

In Table 4, the “Image (text regions removed)” row shows a perception score of 62.22. Given that this benchmark is designed for text-rich scenes, one would expect perception tasks to be highly challenging, if not impossible, without text information. Could you clarify:
1. How was this score achieved despite the absence of text?
2. If accurate, does this suggest that certain benchmark questions may not fully align with the intended text-rich focus?

---

### Official Review · Reviewer_9uor · 2024-11-03

**Soundness:** 2
**Presentation:** 2
**Contribution:** 2
**Rating:** 3
**Confidence:** 4

**Summary:**

The paper introduces MCTBench, a benchmark for evaluating the cognitive abilities of multimodal large language models (MLLMs) in text-rich visual scenes. MCTBench includes two main task types: reasoning tasks for understanding scenes and open-ended content-creation tasks for generating responses. It also incorporates perception tasks to differentiate them from cognitive tasks, minimizing bias from dataset variations.
The benchmark compiles about 5.2k images and 8.5k annotated question-answer pairs across three categories: perception, reasoning, and content creation. Perception and reasoning tasks use multiple-choice formats for easy assessment, while an automated evaluation system, using advanced MLLMs like GPT-4V, is set up for content creation due to the challenges of subjective human evaluation.

**Strengths:**

1）MCTBench provides a thorough assessment of both reasoning and content-creation capabilities in MLLMs, offering a well-rounded evaluation framework.
2）By using advanced MLLMs for automated evaluation, the benchmark reduces the need for costly and subjective human assessments in content creation tasks.
3）By distinguishing between perception and cognitive tasks, MCTBench helps identify specific areas where MLLMs need improvement.The finding that larger models perform better in cognitive tasks provides valuable guidance for future model development and scaling strategies.

**Weaknesses:**

1）Incomplete paper with no content in section 3.3
2）Segmenting cognitive abilities into reasoning and content generation may not be enough, and a sufficiently fine-grained benchmark would require a more precise segmentation of the data
3）Automated evaluations have improved efficiency, but their accuracy and consistency with manual evaluations need further validation

**Questions:**

1) Complete the missing section 3.3 in the paper. What is the specific data construction process in Section 3.3? Can you provide more details about data stratification, annotation, and preprocessing so that other researchers can replicate the MCTBench data preparation process?
2) Can cognitive abilities be further subdivided beyond reasoning and content generation? For example, is it possible to incorporate more refined categories such as logical reasoning, contextual understanding, and cross-modal reasoning to more accurately assess the cognitive abilities of different models?
3) How consistent are automated rating systems with human ratings? Can comparative experiments be conducted to analyze the reliability of automatic scoring in different types of generation tasks and clarify its accuracy in different evaluation dimensions.

---

### Meta-Review · Area_Chair_9ife · 2024-12-20

**Metareview:**

This paper presents MCTBench, a new multimodal benchmark designed to evaluate the cognitive abilities of MLLM through visual reasoning and content generation tasks.

The strengths of this paper include the introduction of an automatic evaluation pipeline to improve the efficiency of the content generation task, and the distinction between perceptual and cognitive tasks to identify specific areas where MLLM needs improvement.

However, the paper is not well formatted, with section 3.3 incomplete and missing text, and figures inserted in the middle of the references section. In addition, the concerns of all reviewers were not addressed due to the lack of a rebuttal by the authors.

Thus, all reviewers gave negative reviews. There is no reason to overturn the decisions of the reviewers.

**Additional Comments On Reviewer Discussion:**

There are no rebuttals and discussions.

---

### Decision · Program_Chairs · 2025-01-22

Reject